# Transcriptome-wide association study for restless legs syndrome identifies new susceptibility genes

Fulya Akçimen [1,2], Faezeh Sarayloo[1,2], Calwing Liao[1,2], Jay P. Ross [1,2], Rachel De Barros Oliveira [2], Patrick A. Dion[2,3] & Guy A. Rouleau [1,2,3 ✉]

Restless legs syndrome (RLS) is a common neurological condition, with a prevalence of 5–15% in Central Europe and North America. Although genome-wide association studies (GWAS) have identified some common risk regions for RLS, the causal genes have yet to be fully elucidated. We conducted a transcriptome-wide association study involving 15,126 RLS cases and 95,725 controls, from the most recent meta-analysis of GWAS, and gene expression weights of GTEx v7 and the CMC dorsolateral prefrontal cortex tissue panels. We identified 13 associations (in 8 independent loci) at the transcriptome-wide significant level, of which 6 were not implicated in the previous GWAS: *SKAP1, SLC36A1, CCDC57, FN3KRP, NCOA6/TRPC4AP*. A fine-mapping approach prioritized *CMTR1, RP1-153P14.5, PRPF6*, and *PPP3R1* – to our knowledge, the latter of which is the first RLS-associated gene directly implicated in dopaminergic pathways. Overall, our findings highlight the power of integrating gene expression data with GWAS to prioritize putative causal genes for functional follow-up studies.

[1] Department of Human Genetics, McGill University, Montréal, QC, Canada. [2] Montreal Neurological Institute and Hospital, McGill University, Montréal, QC, Canada. [3] Department of Neurology and Neurosurgery, McGill University, Montréal, QC, Canada. ✉email: guy.rouleau@mcgill.ca

Restless legs syndrome (RLS) is a sleep-related sensorimotor disorder characterized by an urge to move the legs during periods of rest, especially in the evening and night[1]. The high narrow-sense heritability (~70%) of RLS estimated by twin studies, as well as the strong familial aggregation, have motivated genetic studies of this condition[2,3]. The latest genome-wide association studies (GWAS) meta-analysis identified a total of 19 RLS risk loci that explained 60% of the estimated SNP-based heritability of 19.6% [4]. Using the summary statistics of the most recent RLS-GWAS cohort of European ancestry, various post-GWAS approaches including gene annotation, pathway and gene-set enrichment analyses were applied to prioritize genes in associated loci and identify related biological mechanisms. However, the candidate causal genes at these loci have yet to be completely clarified.

Unlike the previous methods that often associate loci with the nearest gene or focus on individual significant SNP and eQTL associations, transcriptome-wide association studies (TWASs) focus on whole expression and trait associations rather than only top eQTL associations[5]. As a complementary to the previous GWAS, this study aimed to identify novel genes associated with RLS that are not well explained by individual SNP tagging. Through leveraging available transcriptomic imputation approaches, FUSION[5] and S-PrediXcan[6], we sought to integrate eQTL analyses with summary-level GWAS data to determine more detailed information on, and discover novel genes, underlying the pathology of RLS.

## Results

**Study overview.** TWAS was performed using the summary statistics of the largest GWAS meta-analysis at the time of the analysis that contained 15,126 RLS patients and 95,725 control participants in the discovery stage (Fig. 1). Two different approaches, FUSION[5] and S-PrediXcan[6], were applied for transcriptomic imputation using 17 tissue panels and 67,156 gene models (Supplementary Table 1) (see "Methods").

**TWAS genes for RLS.** The top TWAS genes with the strongest associations with a |Z-score| > 3.50, $P < 5 \times 10^{-4}$ are listed in Supplementary Table 2 (with the results from the multi-tissue TWAS presented in Fig. 2). The majority of the identified associations encompass the genes that are in the 1p13.3 (*S100A16*, *S100A2*, *S100A3*, and *S100A3*), 2p14 (*MEIS1* and *PPP3R1*) and 15q23 (*SKOR1*, *MAP2K5*, *IQCH*, *IQCH-AS1*, *AAGAB*, *RP11-34F13.2*, and *TMEM87A*) (Fig. 2).

**Transcriptome-wide significant hits.** A total of seven gene-level models (consisting of five unique genes) reached transcriptome-wide significance ($P = 7.45 \times 10^{-7}$) after Bonferroni correction for 67,156 total tests (Fig. 3). In addition, six significant splicing events were identified using CommonMind Consortium (CMC) dorsolateral prefrontal cortex RNA-seq data (Supplementary Table 3). Among the total 13 significant associations, six of them were not reported in the previous GWAS. These are *SKAP1* and the splicing events that were detected in the *SLC36A1*, *CCDC57*, *FN3KRP*, and *NCOA6/TRPC4AP* genes.

**Fine-mapping prioritized putatively candidate genes for RLS.** A posterior probability of causality for each gene was assigned using the Fine-mapping Of CaUsal gene Sets (FOCUS) software[7]. *CMTR1*, *RP1-153P14.5*, *PPP3R1*, and *PRPF6* were prioritized as putatively candidate genes for RLS with posterior probabilities of 1.00, 0.83, 0.63 and 0.28, respectively (Table 1).

**Gene-set enrichment and pathway analysis.** To identify known biological pathways associated with the top TWAS genes (|Z-score| > 3.50, P < 5E−04), a gene-set enrichment analysis approach was performed using Reactome (https://reactome.org) and Gene ontology (GO; http://geneontology.org). A variety of relevant gene-sets were found to be overrepresented, such as calcium ion binding, metal ion binding, as well as receptor activity pathways (Supplementary Table 4). In addition, putatively causal genes were evaluated for evidence of small molecule druggability or known drugs based on queries of the Drug Gene Interaction database. Potential target molecules were found for *PPP3R1* and *UCKL1* (Supplementary Table 5).

**Phenome-wide association study.** We performed a phenome-wide association study (PheWAS) and identified the risk of other phenotypes for RLS-associated genes, which included metabolic traits such as triglycerides, cholesterol, impedance measures, as well as psychiatric traits such as worrier/anxious feelings, neuroticism, and nervous feelings (Supplementary Data 1 and 2).

## Discussion

This study identified 13 associations at the transcriptome-wide significant level, of which six were not implicated in the previous GWAS. Among these, seven gene-level models (consisting of five unique genes) and six splicing events were found to be associated using CMC dorsolateral prefrontal cortex RNA-seq data. Associations for splicing events highlight an important contribution of variation in RLS risk and suggest an avenue for further functional follow-up studies. However, the direction of the effect should be interpreted with caution since alternatively spliced exons are usually negatively correlated with the risk of a disease[8].

Consistent with previous findings[9,10], expression of three previous RLS GWAS genes, *MEIS1* in the dorsolateral prefrontal cortex and tibial nerve, *SKOR1* in frontal cortex and pituitary, and *MAP2K5* in the dorsolateral prefrontal cortex were associated with RLS. Two additional genes, *IQCH* and *SKAP1*, which were not reported in the previous GWAS, were significantly associated with the RLS.

In the previous RLS GWAS, *MEIS1* was identified as the most significant genetic risk factor, motivating subsequent

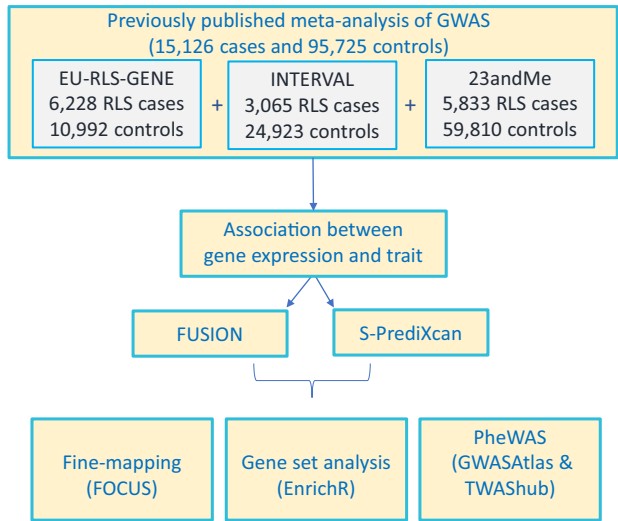

**Fig. 1 Transriptome-wide association study workflow for restless legs syndrome.** A total of 15,126 cases and 95,725 controls of European ancestry from EU-RLS-GENE, INTERVAL, and 23andMe cohorts were used in the discovery stage of the meta-analysis. Transcriptomic imputation was performed using S-PrediXcan and FUSION. A fine-mapping approach was conducted using FOCUS. Gene-set enrichment and phenome-wide association studies were done using EnrichR, GWASAtlas and TWAShub, respectively.

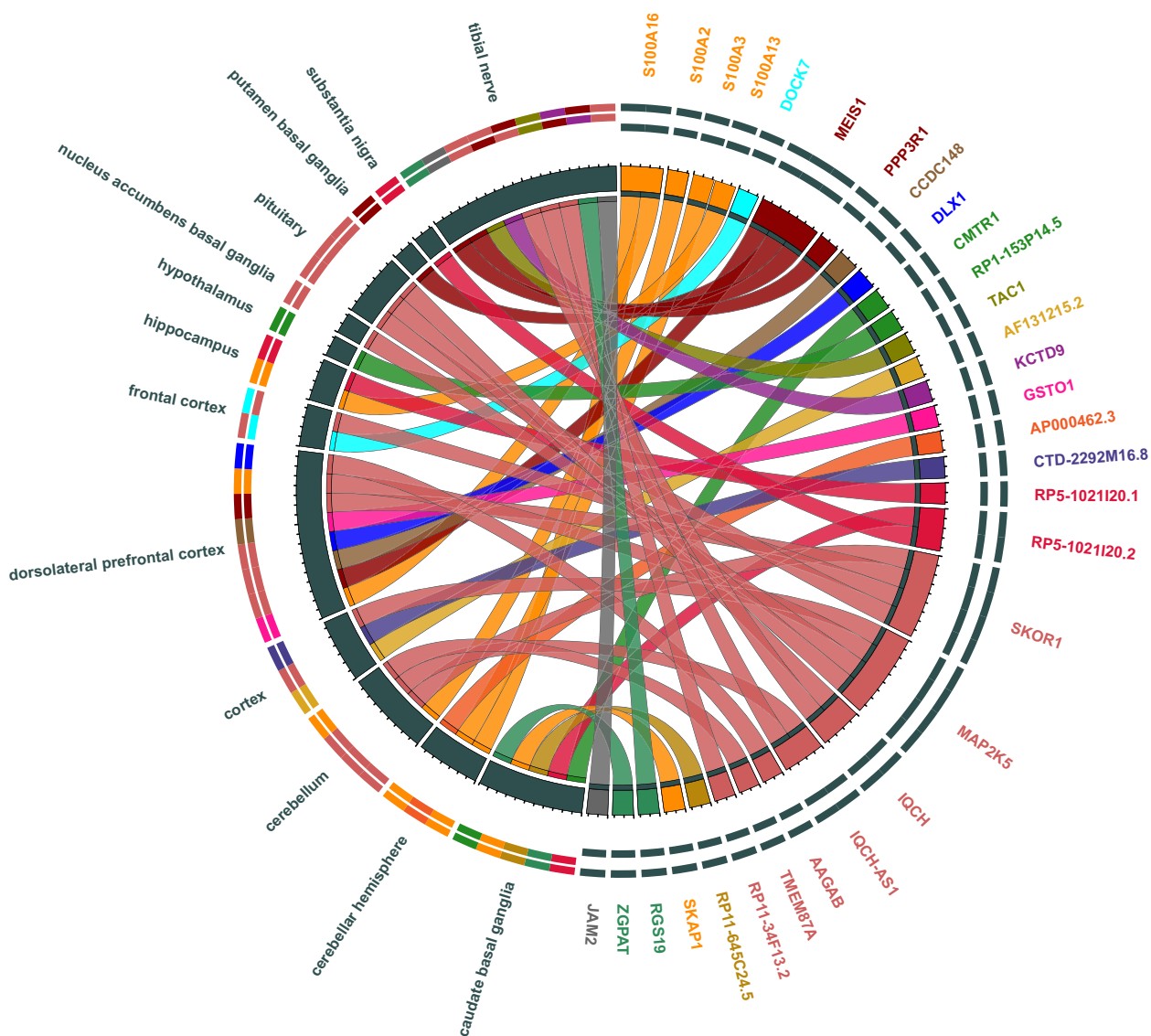

**Fig. 2 Transcriptome-wide association study genes whose differential expression in relevant tissues was associated with restless legs syndrome.**
Ribbons link the genes to the segments that represent the relevant tissues. Gene groups that belong to the different GWAS loci and their relevant tissues were coloured differently. *S100A16, S100A2, S100A3,* and *S100A3*: yellow; *DOCK7*: light blue; *MEIS1* and *PPP3R1*: dark red; *CCDC148*: brown; *DLX1*: dark blue; *CMTR1* and *RP1-153P14.5*: green; *TAC1*: olive; *AF131215.2*: gold; *KCTD9*: purple; *GSTO*: pink; *AP000462.3*: orange; *CTD-2292M16.8*: violet; *RP5-1021I20.1* and *RP5-1021I20.2*: red; *SKOR1, MAP2K5, IQCH, IQCH-AS1, AAGAB, RP11-34F13.2,* and *TMEM87A*: pale pink; *RP11-645C24.5*: khaki; *SKAP1*: goldenrod; *ZGPAT* and *RGS19*: green; *JAM2*: grey. Gene models were visualized using the online version of Circos (http://mkweb.bcgsc.ca/tableviewer/).

functional studies of this gene[9,10]. However, fine-mapping of the corresponding genomic locus prioritized *PPP3R1* with a posterior probability of 0.63 in the 90%-credible gene set. *PPP3R1* encodes for calcineurin subunit B type 1 protein, which functions downstream of dopaminergic pathways[11]. Calcineurin contains iron and zinc in its active site and is regulated by oxidation of an iron cofactor[12,13]. Although the role of altered dopaminergic function and brain-iron homoeostasis in RLS has been well characterized[14,15], none of the previously identified candidate genes were directly implicated as key molecules in dopaminergic neuro-transmission[15]. Being implicated in iron homeostasis[16], as well as expressed mostly in dopamine receptor-positive neurons[11], suggests *PPP3R1* as a candidate gene for the altered brain iron homeostasis and disrupted dopaminergic function leading to the manifestation of RLS.

In summary, by combining the results of two different multi-tissue transcriptomic imputation approaches we have not only helped to clarify candidate genes for associations identified by previous GWAS but also identified several novel genes that might be involved in RLS biology and pathogenesis. This study is the first to impute the expression data and search for associations between gene expression and RLS from summary-level GWAS data. In addition to newly identified RLS-associated genes, several pathways and associated phenotypes that were not reported in the previous post-GWAS analysis were identified using imputed expression data. Although we employed the largest RLS-GWAS cohort of European ancestry, a follow-up study in additional populations would add evidence to support these findings. This study will also provide impetus for further functional research to analyse the consequences of altered gene expression in RLS.

## Methods
**GWAS summary statistics and gene expression data.** Summary statistics of a GWAS meta-analysis consisting of EU-RLS-GENE, INTERVAL[17], and 23andMe cohorts were obtained from Schormair et al.[4] (Fig. 1). A total of 15,126 cases and

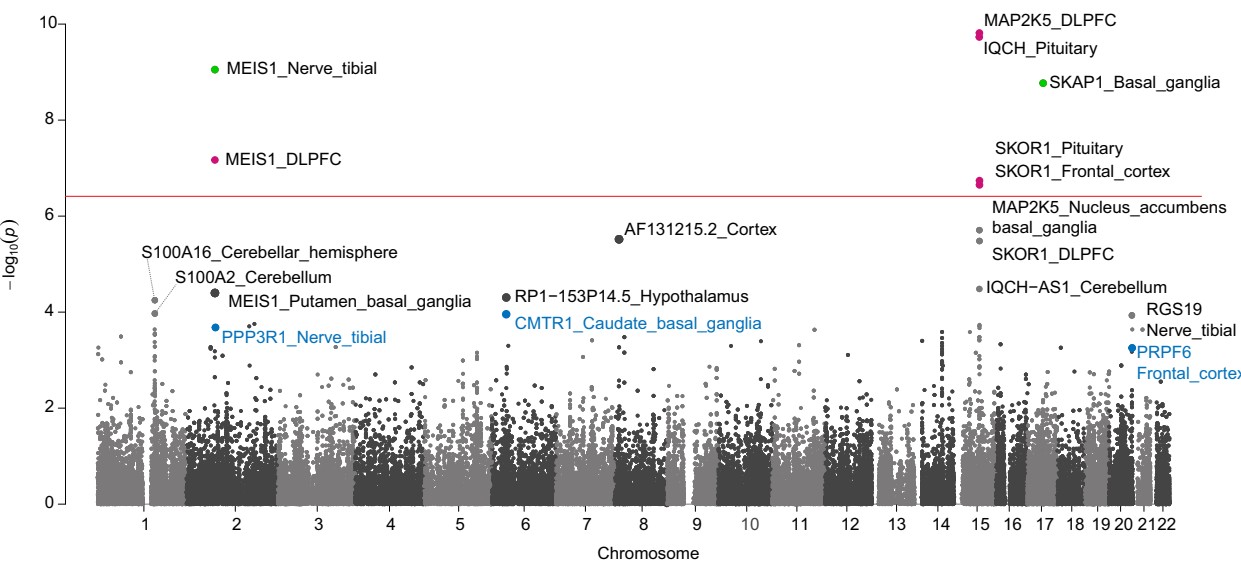

**Fig. 3 Manhattan plot of restless legs syndrome (RLS) transcriptome-wide association study (TWAS) gene models.** Green points represent signals from FUSION, pink represents signals from S-PrediXcan, and blue represents TWAS genes that were prioritized as putative candidates for RLS via statistical fine mapping. A significance threshold of $P = 7.45 \times 10^{-7}$ was used.

**Table 1 Causal posterior probabilities for genes in 90%-credible sets for restless legs syndrome transcriptome-wide association study signals.**

| Gene | Region | Tissue (Z score, software) | Marginal TWAS Z score (FOCUS) | PIP | GWAS nearest gene |
|---|---|---|---|---|---|
| PPP3R1 | 2:68010206-69139564 | Tibial nerve (3.67, S-PrediXcan) | 4.25 | 0.63 | MEIS1 |
| SNRNP27 | 2:68010206-69139564 | Cerebellum (−0.3, S-PrediXcan) | 0.81 | 0.10 | MEIS1 |
| AC019206.1 | 2:68010206-69139564 | Amygdala (−1.91, S-PrediXcan) | −2.26 | 0.06 | MEIS1 |
| TULP1 | 6:35455842-37572375 | Hippocampus (1.46, S-PrediXcan) | 1.83 | <0.01 | BTBD9 |
| ANKS1A | 6:35455842-37572375 | DLPFC (1.56, FUSION) | −1.79 | <0.01 | BTBD9 |
| BRPF3 | 6:35455842-37572375 | Frontal cortex-BA9 (−1.66, S-PrediXcan) | −1.91 | <0.01 | BTBD9 |
| CMTR1 | 6:35455842-37572375 | Caudate-basal ganglia (3.87, S-PrediXcan) | 6.15 | 1.00 | BTBD9 |
| RP1-153P14.5 | 6:35455842-37572375 | Hypothalamus (−4.06, S-PrediXcan) | −4.29 | 0.83 | BTBD9 |
| HOXB3 | 7:45876022-47516523 | Cerebellum (−0.12, S-PrediXcan) | −0.11 | 0.21 | HOXB cluster |
| PRPF6 | 20:62190180-62963102 | Frontal cortex-BA9 (3.4, S-PrediXcan) | 3.32 | 0.28 | MYT1 |
| UCKL1 | 20:62190180-62963102 | Frontal cortex-BA9 (−0.59, S-PrediXcan) | −0.44 | 0.26 | MYT1 |
| STMN3 | 20:62190180-62963102 | Cerebellum (−1.49, S-PrediXcan) | −1.49 | 0.09 | MYT1 |

*BA* Brodmann area 9, *PIP* posterior inclusion probability.

95,725 controls of European ancestry: 6228 and 10,992 from EU-RLS-GENE; 3065 and 24,923 from INTERVAL, and 5833 and 59,810 from 23andMe cohorts were used in the discovery stage of the meta-analysis. The relevant tissue panels from GTEx 53 v7, and the CMC dorsolateral prefrontal cortex were downloaded from the FUSION and PrediXcan websites (Supplementary Table 1). Expression weights of CMC for PrediXcan were obtained from the GitHub (https://github.com/laurahuckins/CMC_DLPFC_prediXcan). It was shown that brain iron homeostasis and dysfunction in the dopaminergic system are involved in the pathogenesis of RLS[18]. Therefore, we used relevant tissue panels of brain and nervous system from GTEx 53 v7, and the CMC dorsolateral prefrontal cortex.

**Transcriptomic imputation.** Transcriptomic imputation was performed using S-PrediXcan and FUSION models applied to genotype data to predict expression values as described in FUSION and S-PrediXcan websites. Both FUSION and S-PrediXcan were performed with default parameters using available expression weights. The European 1000 Genomes v3 LD panel was used for the TWAS. To account for the large number of hypotheses tested, a strict Bonferroni corrected *p*-value threshold of $7.45 \times 10^{-7}$ was used ($\alpha = 0.05/67{,}156$ predictive models). To search for whether the significant TWAS associations were estimated from the same or independent GWAS loci, especially for the associations on the same chromosomes, we checked the linkage disequilibrium between the most significant

GWAS SNPs in the identified loci using LDpair tool (https://ldlink.nci.nih.gov/?tab=ldpair).

**Fine-mapping.** To prioritize genes for each TWAS hits, we performed a gene-based fine-mapping approach using FOCUS. RLS GWAS summary statistics along with eQTL weights were used to assign a posterior probability of causality and estimate a credible set of genes in related tissues. The same 1000 Genomes reference LD panel for FUSION and S-PrediXcan was used as in the analysis described above. A combined eQTL reference panel weight database consisting of GTExv7 weights from PrediXcan and CMC weights from FUSION software was used, as recommended for FOCUS. Finally, the genes in the 90%-credible set with a higher posterior probability were prioritized as putatively causal genes.

**Gene-set enrichment analysis.** A gene-set enrichment analysis approach was conducted using public datasets containing GO (http://geneontology.org) and Reactome (https://reactome.org) pathways in EnrichR[19,20].

**Drug targets.** Genes identified as potentially causal using were interrogated against the gene-drug interactions table of the Drug-Gene Interactions Database (http://www.dgidb.org/). Drugs were mapped to CHEMBL IDs.

**Phenome-wide association studies**. To identify phenotypes associated with the RLS genes identified via TWAS, a phenome-wide association study was conducted using publicly available data provided by GWAS Atlas (https://atlas.ctglab.nl) and TWAS Hub (http://twas-hub.org/). Only the top phenotypes (i.e., those with a $p$-value less than $1 \times 10^{-10}$ in GWAS Atlas or an average chi-square ratio higher than 10 in TWAS Hub) are reported.

**Reporting summary**. Further information on research design is available in the Nature Research Reporting Summary linked to this article.

## Data availability

We declare that the data generated in this study are available within the article and its Supplementary Data files. The summary statistics of the TWAS and full results of PheWAS are included in Supplementary Data 3. Requests for materials relating to the summary statistics of RLS-GWAS meta-analysis that were used in this study should be addressed to EU-RLS-GENE consortium, INTERVAL study (UK) and 23andMe.

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

## Acknowledgements

The authors thank the participants for their contribution to the study. We thank S. Can Akerman for his assistance in preparing the figures. F.A. and C.L. are funded by the Fonds de Recherche du Québec–Santé (FRQS). J.P.R. is funded by the Canadian Institutes of Health Research (CIHR; FRN 159279). G.A.R. holds a Canada Research Chair in Genetics of the Nervous System and the Wilder Penfield Chair in Neurosciences. We thank the International EU-RLS-GENE consortium and the Cooperative Research in the Region of Augsburg (KORA) study for providing RLS GWAS summary statistics. KORA was initiated and is financed by the Helmholtz Zentrum München, which is funded by the German Federal Ministry of Education and Research and by the state of Bavaria. The International EU-RLS-GENE consortium includes data from the COR study which was supported by unrestricted grants to the University of Münster from the German Restless Legs Patient Organisation (RLS Deutsche Restless Legs Vereinigung), the Swiss RLS Patient Association (Schweizerische Restless Legs Selbsthilfegruppe) and a consortium formed by Boeringer Ingelheim Pharma, Mundipharma Research, Neurobiotec, Roche Pharma, UCB (Germany + Switzerland) and Vifor Pharma. The clinical material and biospecimens of the Mayo Clinic Florida RLS collection were collected with the assistance of the Mayo Clinic internal funding through the Neuroscience Focused Research Team grant. Genotyping of the International EU-RLS-GENE consortium dataset was supported by DFG grant 218143125 to Prof. Juliane Winkelmann.

Participants in the INTERVAL randomised controlled trial were recruited with the active collaboration of NHS Blood and Transplant England (www.nhsbt.nhs.uk), which has supported field work and other elements of the trial. DNA extraction and genotyping was co-funded by the National Institute for Health Research (NIHR), the NIHR BioResource (http://bioresource.nihr.ac.uk/) and the NIHR [Cambridge Biomedical Research Centre at the Cambridge University Hospitals NHS Foundation Trust] [*]. The academic coordinating centre for INTERVAL was supported by core funding from: NIHR Blood and Transplant Research Unit in Donor Health and Genomics (NIHR BTRU-2014-10024), UK Medical Research Council (MR/L003120/1), British Heart Foundation (SP/09/002; RG/13/13/30194; RG/18/13/33946) and the NIHR [Cambridge Biomedical Research Centre at the Cambridge University Hospitals NHS Foundation Trust] [*]. A complete list of the investigators and contributors to the INTERVAL trial is provided in ref. [17]. The academic coordinating centre would like to thank blood donor centre staff and blood donors for participating in the INTERVAL trial and Dr Brendan Burchell (University of Cambridge) for advice on RLS phenotyping. *The views expressed are those of the authors and not necessarily those of the NHS, the NIHR or the Department of Health and Social Care.

## Author contributions

F.A. performed all analyses and drafted the manuscript. F.S., C.L., J.P.R. and R.D.B.O. contributed to manuscript writing. P.A.D., and G.A.R. oversaw the manuscript.

## Competing interests

The authors declare no competing interests.
