## [Peer Review File · Communications Biology]

Reviewers' comments:

Reviewer #1 (Remarks to the Author):

Akçimen et al. report a TWAS for restless legs syndrome using published datasets and identify new genes.

Concerns:

The original GWAS from Schormair et al. had already done post-gwas analysis including gene set (pathway) analysis, gene prioritization analysis, genetic correlation with other traits. The authors should make it clear the novelty and strength of this study in the introduction and discussion.

Line 23: The authors report "6 were not implicated in the previous GWAS". This is the main result of this study. Please indicate these genes. Are they new identified gene locus or reported locus with different gene prioritization?

Minor comments:

Line 19: A prevalence of 18% is very high, is it an extreme value? Please report the general prevalence.

Line 21: Please provide the transcriptional data in the abstract and make it clear previous published gwas and expression datasets were used in this study.

Line 51-53: duplicated sentences.

Reviewer #2 (Remarks to the Author):

In this manuscript, the authors describe a transcriptome-wide association study (TWAS) of restless legs syndrome (RLS). They identified 13 gene-trait associations, performed TWAS fine-mapping to pinpoint the possible causal genes, and performed gene set enrichment analysis. Overall, this manuscript could benefit from much more detailed description of their analysis and results.

Major comments:

1) In the abstract, the authors mention that they identified 13 associations at TWAS significant level. It's important to clarify whether these 13 associations are independent signals coming from different LD regions, or correlated signals coming from the same LD regions – in most cases, it's preferable to report the number of independent signals / regions.

2) Additionally, reporting the number of genes might be more informative than reporting the number of associations. How many of the 6 novel associations represent unique genes not discovered in previous GWAS?

3) Line 36: TWAS identifies genes "associated" with complex traits, not just genes that "increase risk of developing complex traits".

4) Line 56: the authors should provide more detail on how they define candidate TWAS genes. What significance threshold was used to select candidate TWAS genes?

5) The authors should provide more detail on the GWAS summary statistics used in their study. More importantly, the authors should clarify whether these GWAS data the authors analyzed are all based on samples of European ancestry.

6) In Methods, the authors mention that they only used 17 tissue panels from GTEx. They authors should provide justifications on why they choose to analyze these 17 tissues, instead of the entire panel, involving 53 tissues. The authors seem to focus only on brain tissues. If that's the case, the authors should discuss reasons for making this choice.

7) Table 1: It would be useful to include chromosome and base pair position information for each gene. Also, it's not clear whether the TWAS Z-score refers to FUSION or S-PrediXcan results.

8) The authors should give more interpretations to Figure 2.

9) Figure 3 is very hard to parse – most of the red, blue, and green dots overlap, making this figure hard to understand. I recommend only coloring the gene models that pass the significance threshold, and leaving the rest of the dots in gray. Also, the author should consider using different color for odd and even chromosomes.

Minor comments:

1) Line 36: what does it mean to "assign genes"?

2) Line 37: the semicolon after "approaches" should be replaced with a comma.

3) There is a typo on line 116.

4) In Supplementary Table 2, the authors should also report the total number of tests, 67156.

We would like to thank the reviewers for their thoughtful review of our paper. Below we respond point by point to the comments made by the reviewers.

Reviewer #1 (Remarks to the Author):

Akçimen et al. report a TWAS for restless legs syndrome using published datasets and identify new genes.

Concerns:

1- The original GWAS from Schormair et al. had already done post-GWAS analysis including gene set (pathway) analysis, gene prioritization analysis, genetic correlation with other traits. The authors should make it clear the novelty and strength of this study in the introduction and discussion.

Response: We would like to thank the reviewer for her/his constructive comments. As recommended, we have discussed the strength and novelty of our approach, comparing to the previous applications used in Schormair et al. Hence the following sentences have been added to introduction and discussion sections of the revised manuscript.

“Using the summary statistics of the most recent meta-analysis of GWAS for RLS, various post-GWAS approaches including gene annotation, pathway and gene-set enrichment analyses were applied to prioritize genes in associated loci and identify related biological mechanisms.”

“Unlike the previous methods that often associate loci with the nearest gene or focus on individual significant SNP and eQTL associations, TWAS focus on the whole cis-SNP associations whether significant or non-significant⁵. As a complementary to the previous GWAS, this study aimed to identify novel genes associated with RLS, that are not well explained by individual SNP tagging.”

“This study is the first to impute the expression data and search for associations between gene expression and RLS from summary-level GWAS data. In addition to newly identified RLS associated genes, several pathways and associated phenotypes that were not

reported in the previous post-GWAS analysis were identified using imputed expression data.“

2- Line 23: The authors report “6 were not implicated in the previous GWAS”. This is the main result of this study. Please indicate these genes. Are they new identified gene locus or reported locus with different gene prioritization?

Response: As recommended, the new genes have been listed in the revised abstract. We also indicated whether they are in newly identified loci or previously reported loci in results.

“We identified 13 associations at the transcriptome-wide significant level, of which 6 were not implicated in the previous GWAS: SKAP1, SLC36A1, CCDC57/FN3KRP, NCOA6/TRPC4AP.”

“Among the total 13 significant associations, 6 of them were not reported in the previous GWAS. These are SKAP1, that was identified in a previously known RLS locus of HOXB cluster, and the splicing events that were detected in the SLC36A1, CCDC57/FN3KRP, NCOA6/TRPC4AP genes.”

Minor comments:

1- Line 19: A prevalence of 18% is very high, is it an extreme value? Please report the general prevalence.

Response: Epidemiological studies have shown that 5–15% of European and North American populations suffer from RLS; in particular, an increased prevalence of up to 18% was reported in the Québec population. As suggested, we reported the general prevalence as follow:

“Restless legs syndrome (RLS) is a common neurological condition, with a prevalence of 5-15% in Central Europe and North America.”

2- Line 21: Please provide the transcriptional data in the abstract and make it clear previous published GWAS and expression datasets were used in this study.

Response: We made it clear that we used the data from previous meta-analysis of GWAS and stated the gene expression datasets in the abstract. These were previously indicated in the methods, as well.

“We conducted a transcriptome-wide association study (TWAS) involving 15,126 RLS cases and 95,725 controls, which were obtained from the most recent meta-analysis of GWAS for RLS, and gene expression weights of GTEx 53 v7 and the CMC dorsolateral prefrontal cortex tissue panels.”

3- Line 51-53: duplicated sentences.

Response: We removed the second sentence.

Reviewer #2 (Remarks to the Author):

In this manuscript, the authors describe a transcriptome-wide association study (TWAS) of restless legs syndrome (RLS). They identified 13 gene-trait associations, performed TWAS fine-mapping to pinpoint the possible causal genes, and performed gene set enrichment analysis. Overall, this manuscript could benefit from much more detailed description of their analysis and results.

Major comments:

1- In the abstract, the authors mention that they identified 13 associations at TWAS significant level. It's important to clarify whether these 13 associations are independent signals coming from different LD regions, or correlated signals coming from the same LD regions – in most cases, it's preferable to report the number of independent signals / regions.

Response: We have clarified it in abstract. There are 13 associations in 7 different loci. Additionally, the genes in the same loci were reported as *CCDC57/FN3KRP*, *NCOA6/TRPC4AP*.

“We identified 13 associations (in 7 independent loci) at the transcriptome-wide significant level, of which 6 were not implicated in the previous GWAS: SKAP1, SLC36A1, CCDC57/FN3KRP, NCOA6/TRPC4AP.”

2- Additionally, reporting the number of genes might be more informative than reporting the number of associations. How many of the 6 novel associations represent unique genes not discovered in previous GWAS?

Response: We reported the number of the unique genes in the results section. Additionally, in the revised manuscript we have indicated the previously reported genes among the 6 novel associations:

“Among the total 13 significant associations, 6 of them were not reported in the previous GWAS. These are SKAP1, that was identified in a previously known RLS locus of HOXB cluster, and the splicing events that were detected in the SLC36A1, CCDC57/FN3KRP, NCOA6/TRPC4AP genes.”

3- Line 36: TWAS identifies genes “associated” with complex traits, not just genes that “increase risk of developing complex traits”.

Response: We thank the reviewer for this correction. We have removed this sentence in the revised paragraph.

4- Line 56: the authors should provide more detail on how they define candidate TWAS genes. What significance threshold was used to select candidate TWAS genes?

Response: This section has been clarified as follow:

“To identify known biological pathways associated with the “top TWAS genes ($|Z\text{-score}| > 3.50, P < 5E-04$)”, a gene-set enrichment analysis approach was performed.”

5- The authors should provide more detail on the GWAS summary statistics used in their study. More importantly, the authors should clarify whether these GWAS data the authors analyzed are all based on samples of European ancestry.

Response: The details on the GWAS data were included in the Fig 1. The GWAS data are based on samples of European ancestry. It was stated in the discussion. In the revised version, we have clarified it in the introduction, as well (line 38).

6- In Methods, the authors mention that they only used 17 tissue panels from GTEx. They authors should provide justifications on why they choose to analyze these 17 tissues, instead of the entire panel, involving 53 tissues. The authors seem to focus only on brain tissues. If that’s the case, the authors should discuss reasons for making this choice.

Response: It has been discussed as followed in introduction section:

“It was shown that brain iron homeostasis and dysfunction in the dopaminergic system are involved in the pathogenesis of RLS⁷. Therefore, we used relevant tissue panels of brain and nervous system from GTEx 53 v7, and the CMC dorsolateral prefrontal cortex.”

7- Table 1: It would be useful to include chromosome and base pair position information for each gene. Also, it’s not clear whether the TWAS Z-score refers to FUSION or S-PrediXcan results.

Response: Table 1 has been revised as suggested. Chromosome and base pair information for each gene, as well as software and Z-score information have been added.

8- The authors should give more interpretations to Figure 2.

Response: More interpretations have been provided in results (lines 58-60).

“The majority of the identified associations encompass the genes that are in the 1p13.3 (S100A16, S100A2, S100A3, S100A3), 2p14 (MEIS1, PPP3R1) and 15q23 (SKOR1, MAP2K5, IQCH, IQCH-AS1, AAGAB, RP11-34F13.2, TMEM87A) (Fig. 2).”

9- Figure 3 is very hard to parse – most of the red, blue, and green dots overlap, making this figure hard to understand. I recommend only coloring the gene models that pass the significance threshold, and leaving the rest of the dots in gray. Also, the author should consider using different color for odd and even chromosomes.

Response: The colors have been modified as suggested.

Minor comments:

1- Line 36: what does it mean to “assign genes”?

Response: We revised this paragraph (lines 42-45). In the revised version, we removed this sentence.

“Unlike the previous methods that often associate loci with the nearest gene or focus on individual significant SNP and eQTL associations, TWAS focus on the whole cis-SNP associations whether significant or non-significant⁵. As a complementary to the previous GWAS, this study aimed to identify novel genes associated with RLS, that are not well explained by individual SNP tagging.”

2- Line 37: the semicolon after “approaches” should be replaced with a comma.

Response: It has been corrected.

3- There is a typo on line 116.

Response: It has been corrected.

4- In Supplementary Table 2, the authors should also report the total number of tests, 67156.

Response: It has been included to Supplementary Table 2.

REVIEWERS' COMMENTS:

Reviewer #1 (Remarks to the Author):

The authors have improved the manuscript and address most of my concerns. I suggest the authors add more details in the Methods section respecting how they defined independent loci.

Reviewer #2 (Remarks to the Author):

I appreciate the authors for addressing my comments. My only remaining major comment is with regards to the interpretation of Figure 2.

Major comment:

1. The authors should provide text to help readers understand what Figure 2 shows. Specifically, in the legend of Figure 2, the authors should clearly explain why different genes are colored differently, and what the curves connecting a gene and tissue represent. Without this information, it's difficult to understand Figure 2.

Minor comments:

1. Line 42 to line 43, the sentence "TWAS focus on the whole cis-SNP associations whether significant or non-significant" is a bit confusing – it's not clear whether the "significant" here refers to significant in GWAS study or eQTL study. I suggest that the authors clarify this ambiguity.

2. I suggest that the authors also provide detailed description (e.g. sample size, ancestry, etc.) of the GWAS in words in the Methods section.

3. I suggest that the authors also discuss reasons why they focus on 17 brain-related tissues in the Methods section.

We would like to thank the reviewers for their thoughtful review of our paper. Below we respond point by point to the comments made by the reviewers.

Reviewer #1 (Remarks to the Author):

The authors have improved the manuscript and address most of my concerns. I suggest the authors add more details in the Methods section respecting how they defined independent loci.

Response: We would like to thank the reviewer for her/his constructive comments. We have checked the linkage disequilibrium between the GWAS loci and revised the manuscript accordingly. The details has been added to the Methods section.

“To search for whether the significant TWAS associations were estimated from the same or independent GWAS loci, especially for the associations on the same chromosomes, we checked the linkage disequilibrium between the most significant GWAS SNPs in the identified loci using LDpair tool (<https://ldlink.nci.nih.gov/?tab=ldpair>).“

Reviewer #2 (Remarks to the Author):

I appreciate the authors for addressing my comments. My only remaining major comment is with regards to the interpretation of Figure 2.

Major comment:

1- The authors should provide text to help readers understand what Figure 2 shows. Specifically, in the legend of Figure 2, the authors should clearly explain why different genes are colored differently, and what the curves connecting a gene and tissue represent. Without this information, it's difficult to understand Figure 2.

Response: We have updated the legend of Figure 2, as recommended.

“Figure 2. Transcriptome-wide association study genes whose differential expression in relevant tissues was associated with restless legs syndrome.

Ribbons link the genes to the segments that represent the relevant tissues. Gene groups that belong to the different GWAS loci and their relevant tissues were colored differently.

S100A16, S100A2, S100A3, and S100A3: yellow; DOCK7: light blue; MEIS1 and PPP3R1: dark red; CCDC148: brown; DLX1: dark blue; CMTR1 and RP1-153P14.5: green; TAC1: olive; AF131215.2: gold; KCTD9: purple; GSTO: pink; AP000462.3: orange; CTD-2292M16.8: violet; RP5-1021I20.1 and RP5-1021I20.2: red; SKOR1, MAP2K5, IQCH, IQCH-AS1, AAGAB, RP11-34F13.2, and TMEM87A: pale pink; RP11-645C24.5: khaki; SKAP1: goldenrod; ZGPAT and RGS19: green; JAM2: grey.

Gene models were visualized using the online version of Circos (<http://mkweb.bcgsc.ca/tableviewer/>).”

Minor comments:

1- Line 42 to line 43, the sentence “TWAS focus on the whole cis-SNP associations whether significant or non-significant” is a bit confusing – it’s not clear whether the “significant” here refers to significant in GWAS study or eQTL study. I suggest that the authors clarify this ambiguity

Response: We thank the reviewer for this concern. To avoid any confusion, we revised this sentence: “*Unlike the previous methods that often associate loci with the nearest gene or focus on individual significant SNP and eQTL associations, transcriptome-wide association studies (TWASs) focus on whole expression and trait associations rather than only top eQTL associations⁵*” (Line 43)

2- I suggest that the authors also provide detailed description (e.g. sample size, ancestry, etc.) of the GWAS in words in the Methods section.

Response: As recommended, these details have been included in words in the Methods. (Lines 133-137)

“Summary statistics of a GWAS meta-analysis consisting of EU-RLS-GENE, INTERVAL17, and 23andMe cohorts were obtained from Schormair et al.⁴ (Fig. 1). A total of 15,126 cases and 95,725 controls of European ancestry: 6,228 and 10,992 from EU-RLS-GENE; 3,065 and 24,923 from INTERVAL, and 5,833 and 59,810 from 23andMe cohorts were used in the discovery stage of the meta-analysis.”

3- I suggest that the authors also discuss reasons why they focus on 17 brain-related tissues in the Methods section.

Response: It has been discussed in the Methods:

“It was shown that brain iron homeostasis and dysfunction in the dopaminergic system are involved in the pathogenesis of RLS¹⁸. Therefore, we used relevant tissue panels of brain and nervous system from GTEx 53 v7, and the CMC dorsolateral prefrontal cortex.”